# Formation and Function of Liquid-Like Viral Factories in Negative-Sense Single-Stranded RNA Virus Infections

**DOI:** 10.3390/v13010126

**Published:** 2021-01-18

**Authors:** Justin M. Su, Maxwell Z. Wilson, Charles E. Samuel, Dzwokai Ma

**Affiliations:** Department of Molecular, Cellular and Developmental Biology & Neuroscience Research Institute, University of California, Santa Barbara, CA 93106, USA; justin.su@lifesci.ucsb.edu (J.M.S.); mzw@ucsb.edu (M.Z.W.); samuel@lifesci.ucsb.edu (C.E.S.)

**Keywords:** viral mediated host remodeling, liquid–liquid phase separation, inclusion body negative-strand RNA virus, viral replication, measles virus, RNA binding protein, membrane, biophysical processes

## Abstract

Liquid–liquid phase separation (LLPS) represents a major physiochemical principle to organize intracellular membrane-less structures. Studies with non-segmented negative-sense (NNS) RNA viruses have uncovered a key role of LLPS in the formation of viral inclusion bodies (IBs), sites of viral protein concentration in the cytoplasm of infected cells. These studies further reveal the structural and functional complexity of viral IB factories and provide a foundation for their future research. Herein, we review the literature leading to the discovery of LLPS-driven formation of IBs in NNS RNA virus-infected cells and the identification of viral scaffold components involved, and then outline important questions and challenges for IB assembly and disassembly. We discuss the functional implications of LLPS in the life cycle of NNS RNA viruses and host responses to infection. Finally, we speculate on the potential mechanisms underlying IB maturation, a phenomenon relevant to many human diseases.

## 1. Introduction

Among all virus taxa, the phylum *Negarnaviricota* embodies all viruses with RNA genomes of negative polarity [1]. This includes viruses with either non-segmented negative-sense (NNS) or segmented negative-sense (SNS) RNA genomes. NNS RNA viruses of order *Mononegavirales* (notably families *Paramyxoviridae, Bornaviridae, Rhabdoviridae, Filoviridae,* and *Pneumoviridae*) and SNS RNA viruses such as Influenza A (family *Orthomyxoviridae*) have historically caused severe human infectious diseases, some with high mortality. These pathogenic viruses have been the center of research efforts to better understand their replication mechanisms at the molecular and cell biological levels [2].

Structure-function studies of NNS and SNS RNA viruses have bridged disciplines between molecular virology and cell biology via elucidating the mechanisms by which viruses replicate. For *Negarnaviricota*, a virion-associated RNA-dependent RNA Polymerase (vRdRP) is the hallmark of viral mRNA transcription and genome replication [1]. Within the nucleocapsid, both NNS and SNS RNA genomes are encapsidated by viral nucleoprotein (N or NP), forming the stable helical ribonucleoprotein (RNP) complex. Structural studies have been conducted to characterize viral RNPs for several NNS RNA viruses, including vesicular stomatitis virus (VSV) [3], rabies virus (RABV) [4], respiratory syncytial virus (RSV) [5], parainfluenza virus type 5 (PIV5) [6], and Ebola virus (EBOV) [7]. Upon infection by these mononegaviruses, viral macromolecular synthesis is cytoplasmic (with exception of *Bornaviridae* where it is nuclear). This is illustrated by the Figure 1 schematic summary of measles virus (MeV) replication. The vRdRP Large (L) polymerase protein with polyribonucleotidyltransferase and methyltransferase activities complexes with its viral processivity factor phosphoprotein (P, VP35 for *Filoviridae*), thereby tethering L to viral RNP for the initiation of de novo RNA synthesis [8,9,10,11]. In contrast, the vRdRP of SNS viruses utilizes a cap-dependent endonuclease to cannibalize host cellular mRNA caps and directly initiates de novo RNA synthesis of each viral RNP segment without the necessity of a phosphoprotein [12,13,14,15]. Collectively, these structural-functional foundations highlight necessary components for viral macromolecular synthesis among the NNS and SNS RNA viruses.

Inspired by theories from soft matter physics, liquid phase condensates are an emerging paradigm for the intracellular organization of membrane-less organelles [16,17,18,19], which differentiates them from classic membrane-bound organelles. Membrane-less and dynamic, these biomolecular condensates compartmentalize diverse structures such as the worm P granules [20], the chromatin [21,22,23], nucleolus [24,25], Cajal bodies [26], PML bodies [27], nuclear speckles [28], P bodies [29], stress granules [30,31], and spindle apparatus [32]. Condensate formation has also been implicated in membrane deformation [33], cargo sorting [34], autophagosome formation [35], transcriptional activation [36,37,38], and stress-induced proteasome-mediated protein degradation in the nucleus [39]. Driven by multivalent macromolecular interactions, these biomolecular condensates are formed via liquid–liquid phase separation (LLPS), with assembly/disassembly regulated by an array of physicochemical factors [16,17,18,19,40]. Notably, biomolecular condensates can adopt a wide range of viscosities: from liquid, to gel, to solid-like states under physiological conditions. For example, unlike the stress granules in human cells that exhibit liquid properties, stress granules in yeast are more solid in nature, presumably the result of adaptation to extreme environments [29]. Similarly, Balbiani bodies of immature oocytes are solid-like structures which likely protect the organelles during oocyte dormancy [41]. On the other hand, the nuclear pore complexes form a gel-like channel which is important for maintaining its permeability barrier [42,43]. Although the functions of these novel phase-separated subcellular structures continue to be elucidated, the primary purposes of these multiphase compartments are thought to increase reaction rates and/or specificities, sequester and/or buffer molecules, and dynamically reorganize molecular hubs [16,17,19].

We have observed the aforementioned paradigm of LLPS extending from cell biological contexts to viral molecular condensates. In retrospect, studies on NNS and SNS RNA viruses played a major role by revealing an unappreciated link between LLPS and virology. Here, we review studies on viral liquid phase-separated cytoplasmic organelles seen among NNS RNA viruses, exploring their formation, functions, and eventual ageing as viral factories. We discuss future directions of study of these liquid organelles and their prospective roles in the viral replication cycle, host responses to infection, and clinical relevance. The LLPS-mediated cytoplasmic condensate has also been shown for Influenza A virus (IAV, a member of SNS RNA viruses) [44]. Compared to NNS RNA viruses, the influenza viral condensate has a different composition (e.g., the lack of a phosphoprotein) and function (e.g., the lack of a role in genomic replication) [45], and thus is not included in the discussion below.

## 2. The Nature and Formation of Viral Liquid-Like Organelles

### 2.1. The Nomenclature

Cytopathic effects from viral infections were well observed in cytological studies and clinical diagnosis [46,47,48], one predominately being the appearance of inclusion bodies (IBs) in the cytoplasm and nucleus. Historically, it has been suggested that either viruses may utilize these inclusions to concentrate viral and host proteins for facilitating viral replication and/or assembly, or the host may orchestrate the inclusion formation in response to viral infection to store viral proteins for degradation [49,50]. As studies progressed, these IBs more commonly became referred to as viral factories, viroplasms, and virosomes [51]. Because the term inclusion bodies is also widely used by biologists and biochemists to describe the insoluble aggregates formed by unfolded or misfolded proteins, it has been proposed to rename these structures or define them based on an identifiable marker [52]. While the field awaits consensus of a new nomenclature, we will adopt the conventional terminology throughout the article.

### 2.2. The Formation

As a hallmark of infection by NNS RNA viruses, the formation of IBs is observed in the cytoplasm of cells infected by paramyxoviruses (e.g., measles virus (MeV) [53,54], mumps virus (MuV) [55], Nipah virus (NiV) [56], parainfluenza virus type 3 (PIV3) [57], and PIV5 [58]), pneumoviruses (e.g., RSV [59] and human metapneumovirus (HMPV) [60]), rhabdoviruses (e.g., RABV [61] and VSV [62]), and filoviruses (e.g., Marburg virus (MARV) [63] and EBOV [64]). These membrane-less supramolecular constructions are spherical when they are small during the early times after infection, and then progress to become heterogeneous in shape and size over time. In many cases, these cytoplasmic inclusion bodies serve as the site of viral replication by concentrating their RNA replication machinery [58,60,61,62,63,64,65].

Biomolecular condensates of proteins and nucleic acids can form a membrane-less organelle via LLPS. Whereas the exact criteria for defining LLPS are still evolving, and additional characteristics have been described, a general consensus in the field is that a LLPS compartment should fulfill the following three criteria: maintaining a spherical shape; fusing with each other after contact; and exchanging molecules with its surrounding environment [66]. Interestingly, several earlier studies including those of EBOV [64] and PIV3 [57] observed the fusion events between small IBs. In addition, Borna disease virus (BDV), a member of the bornaviruses characterized by persistent infection and replication in the nuclei of infected cells, forms nuclear inclusions that actively exchange viral proteins with the surrounding milieu [67]. Based on these observations, together with the fact that viral IBs do not have an enclosing membrane but do contain a high level of viral proteins and RNA as well, it has been tempting to link the formation of IBs to LLPS. Studies on RABV [68], VSV [69], and MeV [70] established that two IBs can fuse and subsequently relax to form a larger spherical IB, displaying hallmarks of surface tension, and that IBs show reversible molecular exchange with the surrounding cytosol after photobleaching. Collectively, these findings established that IBs of these viruses are liquid in nature and phase separate from the cytosol via LLPS.

Whether the formation of IBs with liquid properties represents a universal feature of NNS RNA viruses remains to be determined. Based on the spherical morphology of RSV IBs [52] and the diffusion of N protein observed for the pseudo IBs formed in transfected cells [71], it is very likely that RSV IBs are membrane-less liquid organelles. Moreover, evidence was provided that these IBs are sites of RSV replication. On the other hand, NiV generates two types of IBs, but the viral RNA synthesis appears to occur outside of these structures. One type of IB adopts spherical shape located in the perinuclear region, and the other type has a lamellar or square shape underneath the plasma membrane [56]. These two types of IBs arise independently; covalently-modified viral matrix M protein recruits the nucleocapsid to the plasma membrane and drives the formation of the plasma membrane-associated IBs. Based on these observations, it was proposed that the membrane-associated IBs are sites of virion assembly, whereas the perinuclear IBs are aggresome-like structures [56]. While fusion was detected between pseudo-perinuclear IBs formed in transfected cells, no fusion events have been observed between plasma-membrane associated IBs in both transfected and infected cells. Thus, further studies are needed to determine whether LLPS plays a role in the formation of either IBs formed by NiV.

A transfection approach in the absence of viral infection was used to identify the minimal viral components involved in the formation of RABV, VSV, MeV, and RSV IBs. A common theme that emerged from these results is that co-expression of N and P proteins in transfected cells is either sufficient (for RABV [68], MeV [70], and RSV [71]) or required (for VSV [69]) to generate structures morphologically similar to viral IBs. Moreover, in the case of RABV, MeV, and RSV, co-expression of N and P alone forms IB-like structures that exhibit liquid properties such as fusion/relaxation and/or molecule exchange. These findings were subsequently corroborated and extended by an in vitro reconstitution using MeV [72] and RSV [71] N and P recombinant proteins purified from *E. coli*. The transfection-based and in vitro systems further allow the mapping of the domains within N and P necessary for the IB formation. Not surprisingly, the domain of P mediating its interaction with N (PCTD for RABV, VSV, and RSV; XD for MeV) is required for the formation of IBs. In addition, the dimerization domain (for RABV and VSV) or tetramerization domain (for MeV and RSV) of P protein, and an intrinsic disorder region (IDR) of P (IDD2 for RABV, P_loop_ for MeV, and a.a. (160–227) for RSV), also play an important role. The oligomerization domain and IDR are two common protein structures contributing to the multivalent interactions underlying LLPS [19]. Taken together, these studies demonstrate that the nucleoproteins and phosphoproteins can be viewed as the basic scaffold for formation of the liquid-like IBs during NNS RNA virus infection. Interestingly, whereas expression of the nucleoprotein NP alone seems sufficient for EBOV IB formation with its C-terminal domain being necessary for this process, co-expression with the phosphoprotein VP35 can rescue the deletion of the C-terminal domain of NP to trigger IB formation [73]. Thus, even whether EBOV IBs are liquid organelles remains unknown, VP35 likely plays a role in the formation of IBs in vivo.

Much remains to be learned about the LLPS-driven IB formation. First, although the available evidence indicates nucleoproteins and phosphoproteins are the building blocks of IBs of NNS RNA viruses, we do not know the types of molecular interactions involved. Future studies using the recently developed in vitro reconstitution of MeV or RSV IB formation may provide key insight into this question. In addition to the scaffold proteins, the role of RNA in IB formation has not been extensively interrogated. In this regard, it has been suggested that the formation of RSV pseudo IBs in transfected cells depends on the ability of its N protein to bind RNA and/or oligomerize [71]. It would be interesting to determine whether the NiV peripheral IBs, which do not appear to contain viral RNA [56], are assembled via LLPS. It is also possible that different viruses assemble their IBs using the interactions of different molecular natures.

Second, little is known regarding the host factors that modulate IB assembly/disassembly. Using MeV as a model system, we have found that the IB size is reduced by adding a casein kinase 2 (CK2) inhibitor or mutating the two major CK2 phosphorylation sites of P protein [70]. However, how the phosphorylation status of P modulates the size of MeV IBs is unknown. Since the IDR of MeV N can also be phosphorylated [74], these observations raise the possibility that covalent modifications of N and P may impact MeV IB assembly or disassembly. Other than this, virtually nothing is known about the host factors that can regulate the structure and function of IBs. Given the increasing evidence suggesting a pivotal role of IBs in viral replication, it seems likely that IBs will emerge as hot spots of virus–host interaction. Unfortunately, only a few host proteins have been identified within NNS RNA IBs, partially due to their lack of an enclosing membrane and hence the technical challenge of IB purification. Moreover, host proteins so far identified are seemingly diverse and lack a commonality among different viruses [75]. Proteomic identification of IB-associated host factors (e.g., via the APEX-based [76,77] or BioID-based [78,79] labeling) and characterization of their roles in modulating the structure and function of IBs represent top priorities.

Third, IB-associated granules (IBAGs) were previously described in RSV as a subcompartment inside RSV IBs [65]. This subcompartment contained newly synthetized viral mRNAs and antitermination factor M2-1 (both are required for the IBAG formation), but appeared to exclude structural proteins N, P, L, and the viral genomic RNA. Consistent with the model that M2-1 and viral RNA synthesis are required for IBAG formation, the ectopic co-expression of N and P fails to generate IBAGs in transfected cells [80], but the IBAG-like structure can be observed upon the additional expression of L, M2-1, and a subgenomic replicon [65]. Live-cell imaging shows that IBAGs are liquid-like organelles undergoing dynamic assembly/disassembly cycles, including growth, fusion/relaxation, fission, and disappearance. The presence of IBAGs as a subcompartment of RSV IBs, which have also been suggested as liquid organelles [52], is conceptually similar to the existence of the three liquid compartments within nucleoli (the fibrillar center, the dense fibrillar compartment, and granular compartment) [25]. Future studies are needed to provide insight into the principles governing the formation of immiscible multiphase condensates within IBs. In this context, the localization analysis of viral proteins within IBs can be technically challenging. Immunofluorescence studies on the nucleoproteins and phosphoproteins during infection in the case of several viruses all produce a ring-shaped signal at the surface of the IBs, which is most likely caused by the limited epitope accessibility into the interior IB region [80]. Whereas this obstacle can be overcome by fusing viral proteins to a fluorescent molecule, the results obtained from studies of the fusions should be interpreted with caution due to the potential impacts that the fusion may have on the activity of viral proteins [65]. It is also important to perform live-cell imaging studies to rule out any potential caveats arisen from the fixation process. Meanwhile, the presence of IBAGs in the cases of other NNS RNA viruses has yet to be determined. Newly synthesized RABV RNA forms granules with the viral IBs in fixed cells [81]; these granules exhibit a morphology similar to RSV IBAGs.

## 3. The Function of Viral Liquid-Like Organelles

### 3.1. Implications for the Virus Life Cycle

RNA labeling evidence suggests that cytoplasmic IBs formed in response to infection by RABV [61], VSV [62], EBOV [64], MARV [63], HMPV [60], and RSV [52,65] contain newly synthesized viral RNAs. The PIV5 genomes have also been found in its cytoplasmic IBs [58]. This reinforces the notion that these NNS RNA IBs are bona fide viral factories. Cytoplasmic IBs formed by MeV [53], MuV [55], and PIV3 [57] infection are thought to function in a similar way. Dependent upon the stage of the viral life cycle and viral protein synthesis [82], the vRdRP functions as a transcriptase or a replicase for viral genome RNAs (Figure 1). It has been widely assumed that IBs concentrate both the enzyme (vRdRP) and substrate (the full-length RNP complex) to accelerate viral replication. This idea gained additional support from an in vitro reconstitution study [72]. First, the S491L mutant of MeV N is known to abolish the interaction between the tail domain of N and the XD domain of P and significantly decrease virus production [83]. The same mutant was shown to suppress phase separation in vitro [72]. Second, when synthetic RNA was added to the in vitro system wherein wild-type N was present, either in the dilute phase or the phase-separating condition, the rate of RNA encapsidation was accelerated in the latter, suggesting that LLPS-mediated IB formation can enhance MeV genome replication [72].

Identification of RSV IBAGs indicates that the function of IBs may extend to facilitating viral transcription via a similar mechanism. Since the synthesis of N and P proteins—and hence the primary transcription—is a prerequisite of IB formation, the viral mRNAs that accumulate within the IBAGs presumably arise largely from secondary transcription. An early study with VSV also suggested that viral mRNA synthesis takes place throughout the cytosol during early times post-infection and at late times was subsequently redirected to IBs [62]. Interestingly, the M2-1 protein along with viral mRNA was found released upon the disassembly of RSV IBAGs, suggesting that M2-1 functions as a scaffold for viral mRNAs and directs them toward the cytosol for translation [65]. This model is also consistent with the finding that ribosomal proteins were not present within RSV IBs [65]. These novel observations suggest future directions of study. One concerns the mechanism controlling the assembly and disassembly of RSV IBAGs. It has been shown that both viral RNA synthesis and M2-1 are required for the IBAG formation, but the molecular mechanism is unclear and potentially involves additional proteins (e.g., several translation initiation factors such as eIF4G and PABP were also found to be concentrated within IBAGs [65]). Likewise, the mechanism initiating IBAG disassembly, and thus viral mRNA release, remains a mystery. Another topic for which we have limited understanding is the transport between IBAGs and the cytosol. When IBAGs are embedded with IBs, they contain cytosolic proteins such as eIF4G and PABP. Moreover, viral mRNA released from IBAGs must reach the cytosol to be translated. Thus, how IBAGs exchange materials with the cytosol, in the context of seemingly intact IBs, presents an intriguing question. Lastly, it will be important to determine whether the compartmental segregation represents a general feature of IBs formed by NNS RNA viruses. Since RABV genome does not encode M2-1-like protein, the formation of its putative subcompartments described above [81] may be mediated by other factors yet to be identified.

Whereas the concentration role of IBs would presumably benefit viral transcription and replication, it might also pose a challenge for virion assembly (Figure 1). Considering that the viral envelope glycoproteins reside on the membrane, the RNPs generated within the cytoplasmic IBs need to move from IBs before they then can be transported to the cellular cytoplasmic membrane for the virion assembly. In this regard, ejections of punctate structures, presumably corresponding to condensed RNPs, from the RABV IB have been visualized by live cell imaging [68], but the mechanism remains unknown. In principle, the presumptive RNP ejection could be a result of dynamic equilibrium of RNPs between IBs and their surrounding environment, or it could be triggered under a specific condition. The observation that no ejection was detected in the minimal system (i.e., cells co-transfected with N and P) strongly suggests that additional viral or host components are required for RNP ejections, possibly by regulating the physiochemical property of the RABV IBs. How RNPs produced in the IBs of other NNS RNA viruses are assembled into the virus particles remains another open avenue for future investigations.

### 3.2. Implications for Host Responses to Infection

Innate immune responses, the cornerstone of which is the interferon response, represent the first line of defense against viral pathogens [84,85,86,87]. In some instances, this system detects evolutionary conserved features of pathogens, known as pathogen-associated molecular patterns (PAMPs) via a variety of pattern recognition receptors (PRRs). Activation of PRRs subsequently leads to activation of interferon and inflammatory response. In the case of RNA viruses, two PRRs RIG-1 and MDA5 [86,87] recognize double-stranded (ds) RNA in the cytosol, and signal via the MAVS mitochondrial adaptor to activate the type 1 IFN and NF-κB signaling pathways [88,89,90]. Whereas previous reviews have documented how RNA viruses have evolved methods to negate these host antiviral pathways [91], few of the studies addressed the role of IBs in these processes, and most of our knowledge in this aspect is derived from that of RSV. Based on immunofluorescence evidence, the sequestration of MAPK p38 and OGT into RSV IBs was described [92]. It was proposed that the sequestration suppresses MK2 activity and stress granule assembly, which may benefit the virus by inhibiting innate responses [92]. Utilizing a similar immunofluorescence approach, both MDA5 and MAVS [80], and most recently the p65 subunit of NF-κB [52], have also been shown to become sequestered into IBs following RSV infection. Consistently, the level of induced IFNβ mRNA [80] and the nuclear translocation of NF-κB [52] were greatly diminished. Together, these studies raise questions for future investigation. First, the mechanism of sequestration remains elusive. Although the MDA5/MAVS and NF-κB p65 association with IBs in cells ectopically expressing N and P favors a model whereby N and/or P proteins can recruit these innate immunity signaling components to IBs as a complex, the immunostaining patterns between N/P and the immune modulators are different within IBs. To explain the differential localization, it was proposed that the initial recruitment of p65 to IBs may require the low affinity interactions with N and/or P, and that the subsequent maintenance of its IB localization is enhanced by other multivalent interactions within IBs [52]. A second intriguing topic involves the contributions of sequestration to the suppression of innate immunity. Although not yet quantitated, viral IBs, and especially their subcompartments, probably only account for a small fraction of the total cytosol volume. In order to effectively block NF-κB signaling—and by analogy, MDA5/MAVS signaling—either the great majority of p65 must be rerouted into the small volume fraction occupied by the corresponding IB subcompartment soon after its release from IκB, or alternatively the activity of p65 may be inhibited in another manner independent from p65 entrapment within IBs [52]. A more detailed quantification of the fraction of total p65 associated with IBs will provide a platform for dissecting the role of IBs in sequestration of innate response components. A third emerging topic relates to the aforementioned concept of subcompartments. It was reported that the p65 immunofluorescence signal does not co-localize with those of IBAG markers (i.e., M2-1 and viral RNA); this then led to the hypothesis that there are multiple subcompartments within IBs [52]. Finally, in addition to their roles in innate immunity, IBs are conceivably designated sites of interplay between viral infection and other host signaling pathways, such as those associated with PKR-mediated stress response [93,94].

### 3.3. Maturation of the Viral IBs

In vitro IDR-based condensates have the propensity to undergo maturation, also known as molecular aging, by transitioning from a liquid state to a gel or solid-like state at higher protein concentrations and/or over time [17]. The gel or solid-like condensates have also been documented in vivo, suggesting that the phase maturation can also occur under a physiological setting [29,41]. Indeed, there is growing evidence that the abnormally-formed solid-like states of biomolecular condensates are linked to disease, especially neurodegeneration [40]. Consistent with the essential role of IDRs in driving viral IB formation, we have found that the large MeV IBs display a slower and incomplete molecular exchange with the surrounding soluble phase in the FRAP analysis [70], indicating a gel-like state. Similar conclusions have been made for the large IAV [44] and RSV [52] IBs based on their resistance to hypotonic shock. These observations suggest that IB maturation might represent a general feature among NNS and SNS RNA viruses.

What causes IB maturation remains an open question. Maturation may result from the passive biophysical effect caused by the increased number of IDR-IDR interactions of IDR-containing nucleoprotein and phosphoprotein, leading to entanglement or vitrification and possibly the formation of fibrous structures. Alternatively, maturation could be a sign of active viral and/or host processes (e.g., covalent modifications) that modify the valency or interaction strength of proteins in the IBs and result in higher viscosity and lower dynamic exchange with the cytosol. Given that both N [74] and P [95,96] can be phosphorylated within their IDRs and phosphorylation of MeV P affects IB size [70], one could examine the role of P and/or N phosphorylation in IB maturation using the in-vitro reconstitution system.

Another intriguing possibility is that IB maturation may be triggered by the altered environmental conditions as a result of viral replication activity. Active viral transcription and replication within IBs would be expected to energetically tax the cell, and hence require a high level of ATP. Indeed, cells infected with hepatitis C virus (HCV), a positive-sense single-stranded RNA viruses, display a lower overall cytoplasmic ATP level [97]. Moreover, visualization and measurement of ATP levels in HCV-infected live cells have revealed an elevated level (up to fivefold) of ATP at the putative sites of viral replication compared to their surroundings [97]. In addition to supporting viral replication, increased local ATP levels may function as a biological hydrotrope [98] to increase protein solubility within IBs. Alternatively, elevated ATP may be needed to energize the cell machinery to keep the condensates in the liquid state, as observed for stress granules [99]. It is conceivable that as the viral replication enters the late stage, the cellular ATP level becomes sufficiently depleted, and hence large IBs will become prone to maturation because of their high contents of protein and RNA. Interestingly, African swine fever virus, a member of the large DNA virus *Asfarviridae* family known to replicate in the cytoplasm, recruits mitochondria to its IBs [100]. The proximity of mitochondria to RSV IBs has also been reported [80]. Future studies are needed to determine whether ATP has a role in viral IB maturation and whether the recruitment of mitochondria to the proximity of IBs facilitates the delivery of ATP produced in mitochondria to IBs. Moreover, the possibility that IB maturation is induced by other environmental perturbations cannot be excluded. Regardless of the mechanism involved, these studies will advance our general understanding of maturation of biomolecular condensates. They may also provide more insight into the formation of those protein aggregates associated with neurodegenerative diseases that are solid-like amyloid fibrils characterized by cross β-strand interactions [101,102].

The functional implication of IB maturation in the viral life cycle warrants examination as well. Since the large IBs can only be detected during the late stage of infection, one possibility is that these large IBs can better support viral replication. However, this possibility is seemingly at odds with the observation that all RSV IBs contain newly synthesized viral RNAs and genomic RNA [65]. Another possibility is that the IB size may regulate the assembly and budding of viral RNPs due to the size-dependent physiochemical properties. One can test this hypothesis by tracking and comparing the RNP ejections [68] from IBs of different sizes. Additionally, it would be interesting to examine whether the matrix protein even makes transient contact with IBs; if so, then whether the frequency of such contact correlates to the IB size. A further possibility is that large IBs are more efficient at sequestering and/or excluding host antiviral factors. There have been cases of physiological condensates requiring gel-like states for their functions. For example, the nuclear pore complex acts as a diffusion barrier for proteins above 40 kDa by forming a phase separated gel-like structure in the pore [103]. This model is consistent with the observation that NF-κB p65 can only be detected in large RSV IBs [52]. Super-resolution microscopy may provide a more definitive test for this hypothesis since it remains possible that the p65 granules associated with small IBs are too small to be detected by conventional fluorescent microscopy.

## 4. Perspectives

Numerous studies during the past decade have established that LLPS represents an important principle for organizing cellular membrane-less compartments known as biomolecular condensates. Upon infection, NNS RNA viruses including measles virus (Figure 1) form IBs in the cytoplasm of host cells. Whereas these IBs are known to concentrate a subset of viral proteins, especially those involved in viral replication, they do not have an enclosing membrane. As we discussed, this observation had led the recent discovery that these viruses (notably RABV, VSV, and RSV in addition to MeV) employ LLPS as a strategy to assemble IBs. A liquid-like cytoplasmic organelle was also described in the case of IAV [44], a member of SNS RNA viruses. Since then, LLPS of viral components has been shown in a variety of other viruses including SARS-CoV-2 (a positive-sense RNA virus) [104,105], HIV-1 (a retrovirus) [106], and HSV-1 (a dsDNA virus) [107]. Thus, LLPS-driven assembly of IBs is likely to be a pan-viral phenomenon. Although the number of reports addressing the role of LLPS in viral infection have been rapidly growing, phase separation research in virology is still in its infancy with many important questions remaining to be answered.

From the structural perspective, there is strong evidence that nucleoproteins and phosphoproteins are two key scaffold components of IBs formed by NNS RNA viruses. However, the role of RNA, in particular viral genomic RNA, in the phase separation remains to be explored. We also need to determine how a subcompartment is formed and maintained within IBs as well as the multiplicity and rearrangement of subcompartments possibly within an IB (e.g., see [108]). Virtually nothing is known about the host factors which regulate the assembly, maturation, and disassembly of IBs and their subcompartments. From the functional perspective, the data available suggest that viral IBs can promote viral RNA synthesis, presumably via concentrating the viral machinery; however, the definitive demonstration of this hypothesis would require the development of a strategy that allows us to specifically disrupt IB formation in infected cells. Moreover, little investigation has been conducted regarding the mechanism linking IBs to virion assembly. If the occurrence of subcompartments becomes a general theme, then elucidating how compartmental activities are coordinated within the hub of IBs becomes warranted. Finally, studies using RSV have shown that IBs can function to sequester the innate immunity signaling molecules. It is important to discern whether IBs formed with other viruses confer similar functions. The potential relationship between IBs and other stress signaling pathways, or other cellular structures and organelles, also represents another area of needed investigation. We expect that in vitro reconstitutions (only achieved recently in MeV); the proteomic characterization of IBs; the high-resolution imaging techniques; and the development of new tools to control IB assembly/disassembly in cells will collectively contribute much to our further understanding of these well-established viral factories.

Given the need for additional antiviral therapeutics, studies on viral IB are relevant to human health. If it were possible to target viral IBs without negatively impacting other physiological condensates within a cell, we may significantly alleviate the pathological effects of viral infection. Thus, the clinical implications behind viral IBs are significant, and may open promising paths of translational research. Distinctly, the IB scaffolds of NNS RNA viruses, nucleoproteins and/or phosphoproteins, contain a high content of IDRs [109]. Recent bioinformatics analyses of viral proteins further revealed that many of the viral IDRs are prion-like disordered domains (PrLD) [110]. PrLDs are intimately linked to many neurodegenerative diseases because they are prone to misfold and form insoluble solid-like protein aggregates via aberrant phase transition [111]. Characterization of the molecular mechanism governing viral IB maturation may provide novel insight into the formation of neurodegenerative plaques.

## Figures and Tables

**Figure 1 viruses-13-00126-f001:**
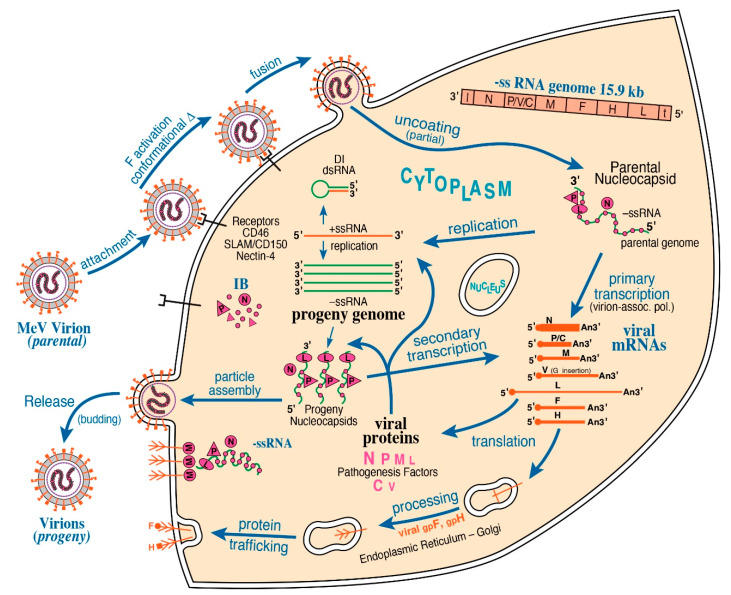
Schematic diagram of measles virus (MeV) multiplication cycle. MeV possesses a negative-sense, single-stranded RNA genome of ~15.9 kb. Virus multiplication occurs in the cytoplasm. Following receptor-mediated virion attachment, the viral envelope fuses with host membranes, releasing the ribonucleocapsid (RNP) complex consisting of genome RNA encapsidated by nucleoprotein N into the cytoplasm; this triggers activation of the virion-associated RNA polymerase complex of viral L and P proteins. The RNP constitutes the basic machinery responsible for RNA synthesis, initially for transcription and then for replication. These processes are thought to be associated with the cytoplasmic inclusion body (IB) factories. Viral C protein interacts with P and enhances polymerase processivity; in the absence of C, defective interfering (DI) RNAs are readily formed. IB assembly triggered by MeV infection is mediated by liquid–liquid phase separation (LLPS). MeV N and P proteins are sufficient to form IB structures by LLPS. The viral matrix M protein mediates assembly of progeny enveloped virions that include two viral glycoproteins, the hemagglutinin H attachment protein and the F fusion protein.

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
