# Peer review of "Formation and Function of Liquid-Like Viral Factories in Negative-Sense Single-Stranded RNA Virus Infections"

_viruses, 2021, doi:10.3390/v13010126_

Round 1
Reviewer 1 Report
This is a well-written and timely review considering an interesting and important subject, roles of the liquid-like viral factories in negative-sense single-stranded RNA viruses. The manuscript is of utmost importance and will have a noticeable impact.
Author Response
We appreciate the reviewer’s positive comment regarding the importance and impact of our manuscript.
Reviewer 2 Report
Recent results published by different teams have shown that the viral factories of several single-stranded negative RNA viruses were membraneless organelles formed by liquid liquid phase separation. The revelation of the nature of viral factories allows to consider the functioning and regulation of these objects from a new angle. Su et al present here a review of the evidence in favor of the "liquid" nature of the viral factories of single-stranded negative RNA viruses and the few data available on the viral elements allowing the assembly of these biocondensates. They describe the functions associated with these compartments knowing that the data linking the nature of viral factories to their functions remain extremely limited today. In this very comprehensive and easy-to-read review, the authors insist on the many questions raised by the nature of viral factories that remain unanswered today.
I have a few minor comments.
Lines 25-30: The full description of the taxonomy is tedious, especially since segmented negative RNA viruses are not the subject of the review.
It would be desirable to clarify the relationship between Haploviricotina and Mononegavirales?
Lines 48-52: The data for SNS RNA viruses are a bit too detailed as they are not the subject of the review.
Line 110: Jobe et al, refer to IBs as "viral inclusion bodies" but it does not seem to me that they propose a new nomenclature.
Line 117 : Reference 59 is not the most adapted. This one could be better doi: 10.1006/viro.1993.1366
Line 152-172: Galloux et al, described the P domains of RSV involved in the formation of IBs and the in vitro reconstitution of condensates from purified N and P (doi: 10.1128/mBio.01202-20.). These results should be referred to in this paragraph (and L141 and may be somewhere else)
Lines 221-222: Most of the articles cited show that IBs not only contain viral RNAs but are the site of synthesis of these RNAs.
Lines 228-231: Why is the fact that a measles mutated N that blocks the interaction between N and P inhibits the formation of biocondensates, an argument in favor of accelerating the encapsidation of viral RNA in viral factories.
Lines 342-344: It is not clear why the association of mitochondria with viral factories suggests a role of ATP depletion in the maturation of viral factories.
Author Response
We thank the reviewer for his/her helpful comments. We are pleased that the reviewer found the manuscript a very comprehensive and easy-to-read review. Our point-to-point responses below summarize how we addressed the reviewer’s comments that overall were minor.
Lines 25-30: The full description of the taxonomy is tedious, especially since segmented negative RNA viruses are not the subject of the review.
Response: Thanks. We shortened the taxonomy description (now line 26-27) following the reviewer’s suggestion.
It would be desirable to clarify the relationship between Haploviricotina and Mononegavirales?
Response: The term “Haploviricotina” has been removed.
Lines 48-52: The data for SNS RNA viruses are a bit too detailed as they are not the subject of the review.
Response: Thanks. We shortened the description (now line 47-49) following the reviewer’s suggestion.
Line 110: Jobe et al, refer to IBs as "viral inclusion bodies" but it does not seem to me that they propose a new nomenclature.
Response: Thanks. We modified the sentence as “it has been proposed to rename these structures or define them based on an identifiable marker” (now line 108-109).
Line 117: Reference 59 is not the most adapted. This one could be better doi: 10.1006/viro.1993.1366
Response: We thank the reviewer for his/her suggestion and have changed the citation accordingly (now line 115).
Line 152-172: Galloux et al, described the P domains of RSV involved in the formation of IBs and the in vitro reconstitution of condensates from purified N and P (doi: 10.1128/mBio.01202-20.). These results should be referred to in this paragraph (and L141 and may be somewhere else)
Response: We thank the reviewer for pointing this out. We have added the above reference and discussed its findings at several places, including line 137-139, line 151-164, line 175, and line 177-178.
Lines 221-222: Most of the articles cited show that IBs not only contain viral RNAs but are the site of synthesis of these RNAs.
Response: We have modified the text as “[…] contain newly synthesized viral RNAs.”
Lines 228-231: Why is the fact that a measles mutated N that blocks the interaction between N and P inhibits the formation of biocondensates, an argument in favor of accelerating the encapsidation of viral RNA in viral factories.
Response: Unlike the encapsidation experiment, the experiment using the N(S491L) mutant was conducted in the absence of RNA. To help avoid potential confusion, we have specified that the wild-type N is used for the encapsidation study (line 234).
Lines 342-344: It is not clear why the association of mitochondria with viral factories suggests a role of ATP depletion in the maturation of viral factories.
Response: The speculation for a possible role of ATP depletion in IB maturation is primarily based on the observations that ATP can function as a biological hydrotrope to increase the protein solubility. ATP can energize the cellular machinery to keep the biocondensates in the liquid state and viral replication can lead to a reduction of overall cytoplasmic ATP. On the other hand, the recruitment of mitochondria to the proximity of IBs may facilitate the delivery of ATP produced by mitochondria to IBs (line 346-348).